# NEXT BLOCK PREDICTION: VIDEO GENERATION VIA SEMI-AUTO-REGRESSIVE MODELING

## ABSTRACT

Next-Token Prediction (NTP) is a de facto approach for autoregressive (AR) video generation, but it suffers from suboptimal unidirectional dependencies and slow inference speed. In this work, we propose a semi-autoregressive (semi-AR) framework, called **N**ext-**B**lock **P**rediction (NBP), for video generation. By uniformly decomposing video content into equal-sized blocks (e.g., rows or frames), we shift the generation unit from individual tokens to blocks, allowing each token in the current block to simultaneously predict the corresponding token in the next block. Unlike traditional AR modeling, our framework employs bidirectional attention within each block, enabling tokens to capture more robust spatial dependencies. By predicting multiple tokens in parallel, NBP models significantly reduce the number of generation steps, leading to faster and more efficient inference. Our model achieves FVD scores of 55.0 on UCF101 and 25.5 on K600, outperforming the vanilla NTP model by an average of 4.4. Furthermore, thanks to the reduced number of inference steps, the NBP model generates 8.89 frames ($128 \times 128$ resolution) per second, achieving an 11× speedup in inference. We also explored model scales ranging from 700M to 3B parameters, observing significant improvements in generation quality, with FVD scores dropping from 25.5 to 19.5 on K600, demonstrating the scalability of our approach.

## 1 INTRODUCTION

The advance of Large Language Models (LLMs) such as ChatGPT (OpenAI, 2023), GPT-4 (Achiam et al., 2023) and LLaMA (Touvron et al., 2023) has cemented the preeminence of Auto-Regressive (AR) modeling in the realm of natural language processing (NLP). This AR modeling approach, combined with the decoder-only Transformer architecture (Vaswani et al., 2017), has been pivotal in achieving advanced levels of linguistic understanding, generation, and reasoning (Wei et al., 2022; OpenAI, 2024a; Chen et al.). Recently, there is a growing interest in extending AR modeling from language to other modalities, such as images and videos, to develop a unified multimodal framework (OpenAI, 2024b; Team, 2024; Lu et al., 2023; Wu et al., 2023). This extension brings numerous benefits: (1) It allows for the utilization of the well-established infrastructure and techniques from the LLM community (Dao et al., 2022); (2) The scalability and generalizability of AR modeling, empirically validated in LLMs (Kaplan et al., 2020; Yu et al., 2023a), can be extended to the multimodal domains to strengthen models (Henighan et al., 2020); (3) Cognitive abilities observed in LLMs can be transferred and potentially amplified with multimodal data, moving closer to the goal of artificial general intelligence (Bubeck et al., 2023).

Given the inherently autoregressive nature of video data in temporal dimensions, video generation is a natural area for extending AR modeling. Vanilla AR methods for video generation typically follows the Next-Token Prediction (NTP) approach, i.e., tokenize video into discrete tokens, then predict each subsequent token based on the previous ones. However, this approach has notable limitations. First, the generation order of NTP often follows a unidirectional raster-scan pattern (Hong et al., 2023; Wang et al., 2024; Yan et al., 2021), which fails to capture strong 2D correlations within video frames, limiting the modeling of spatial dependencies (Tian et al., 2024). Second, NTP necessitates a significant number of forward passes during inference (e.g., 1024 steps to generate a 16-frame clip), which reduces efficiency and increases the risk of error propagation (Bengio et al., 2015).

In this work, we propose a semi-autoregressive (semi-AR) framework, called **N**ext-**B**lock **P**rediction (NBP), for video generation. To better model local spatial dependencies and improve inference efficiency, our framework shifts the generation unit from individual tokens to blocks (e.g., rows or frames). The objective is also redefined from next-token to next-block prediction, where each token in the current block simultaneously predicts the corresponding token in the next block. In contrast to the vanilla AR framework, which attends solely to prefix tokens, our NBP approach allows tokens to attend to all tokens within the same block via bidirectional attention, thus capturing more robust spatial relationships. By predicting multiple tokens in parallel, NBP models significantly reduce the number of generation steps, resulting in faster and more computationally efficient inference.

Experimental results on the UCF-101 (Soomro et al., 2012) and Kinetics-600 (K600) (Carreira et al., 2018) datasets demonstrate the superiority of our semi-AR framework. With the same model size (700M parameters), NBP achieves a 25.5 FVD on K600, surpassing the vanilla NTP model by 4.4. Additionally, due to the reduced number of inference steps, NBP models can generate 8.89 frames (128×128 resolution) per second, achieving an 11× speedup in inference. Compared to previous state-of-the-art token-based models, our approach proves to be the most effective. Scaling experiments with models ranging from 700M to 3B parameters show a significant improvement in generation quality, with the FVD score dropping from 25.5 to 19.5, highlighting the scalability of the framework. We hope this work inspires further advancements in the field.

## 2 RELATED WORK

**Video Generation.** Prevalent video generation frameworks in recent years include Generative Adversarial Networks (GANs) (Yu et al., 2022; Skorokhodov et al., 2021), diffusion models (Ho et al., 2022; Ge et al., 2023; Gupta et al., 2023; Yang et al., 2024), auto-regressive models (Hong et al., 2023; Yan et al., 2021; Kondratyuk et al., 2023), etc. GANs can generate videos with rich details and high visual realism, but their training is often unstable and prone to mode collapse. In contrast, diffusion models exhibit more stable training processes and typically produce results with greater consistency and diversity (Yang et al., 2022). Nevertheless, AR models demonstrate significant potential for processing multi-modal data (e.g., text, images, audio, and video) within a unified framework, offering strong scalability and generalizability. To align with the trend of natively multimodal development (OpenAI, 2024b), this paper focuses on exploring video generation using AR modeling.

**Auto-regressive Models for Video Generation.** With the success of the GPT series models (Brown et al., 2020), a range of studies has applied AR modeling to both image (Chen et al., 2020; Lee et al., 2022) and video generation (Hong et al., 2023; Wang et al., 2024; Yan et al., 2021). For image generation, traditional methods divide an image into a sequence of tokens following a raster-scan order and then predict each subsequent token based on the preceding ones. In video generation, this process is extended frame by frame to produce temporally-coherence content. However, conventional AR models predict only one token at a time, resulting in a large number of forward steps during inference. This significantly impairs the generation speed, especially for high-resolution images or videos containing numerous tokens (Liu et al., 2024).

**Semi-Auto-regressive Models.** To improve the efficiency of AR models, researchers in the NLP field have explored speculative decoding (Xia et al., 2023) and parallel decoding (Stern et al., 2018) algorithms. These methods typically use multiple output heads or modules to predict several future tokens based on the last generated token (Gu et al., 2017; Gloeckle et al., 2024). Given that video content can be uniformly decomposed into blocks of equal size (e.g., row by row or frame by frame), we propose a framework where each token in the last block predicts the corresponding token in the next block, without requiring additional heads or modules. Recent research in the image generation field has also revisited the token generation order in AR models, leading to faster generation processes. For example, VAR (Tian et al., 2024) generates 2D token maps progressively from coarse to fine scales, while MAR (Li et al., 2024) predicts multiple tokens simultaneously in a randomized order using special `[MASK]` tokens. Compared to VAR, our method decomposes visual inputs into spatio-temporal blocks rather than across multiple resolution scales, resulting in more than 2× shorter token sequences [1] and improved inference efficiency for video generation. In contrast to MAR, our

---

[1]Our method uses an average of 256 tokens to represent a $256 \times 256$ frame, while VAR requires 680 tokens.

approach eliminates the need for mask token modeling, providing a denser supervised signal and higher training efficiency.

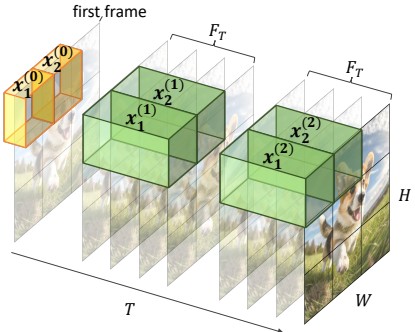

Figure 1: 3D discrete token map produced by our video tokenizer. The input video consists of one initial frame, followed by $n$ clips, with each clip containing $F_T$ frames. $x_j^{(i)}$ indicates the $j^{th}$ video token in the $i^{th}$ clip.

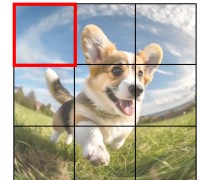 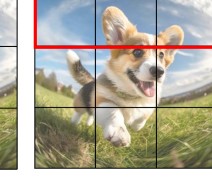 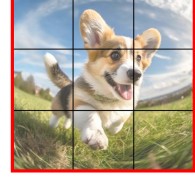

Block size = **1x1x1**   Block size = **1x1x3**   Block size = **1x3x3**
(token-wise, vanilla AR)   (row-wise)   (frame-wise)

Figure 2: The three examples of block include token-wise, row-wise, and frame-wise representations. When the block size is set to $1\times1\times1$, it degenerates into a token, as used in vanilla AR modeling. Note that the actual token corresponds to a 3D cube, we omit the time dimension here for clarity.

## 3 METHOD

In this section, we first introduce our video tokenizer § 3.1, highlighting its two key features: joint image-video tokenization and temporal causality, both of which facilitate our semi-AR modeling approach. Next, we provide a detailed comparison between vanilla Next-Token Prediction (NTP) (§ 3.2) and our **N**ext-**B**lock **P**rediction (NBP) modeling (§ 3.3). Our NBP framework employs a block-wise objective function and attention masking, enabling more efficient capture of spatial dependencies and significantly improving inference speed.

### 3.1 VIDEO TOKENIZATION

We utilize MAGVITv2 Yu et al. (2024) as our video tokenizer, which is based on a causal 3D CNN architecture. Given a video $\mathbf{X} \in \mathbb{R}^{T \times H \times W \times 3}$ in RGB space,[2] MAGVITv2 encodes it into a feature map $\mathbf{Z} \in \mathbb{R}^{T' \times H' \times W' \times d}$, where $(T', H', W')$ is the latent size of $\mathbf{Z}$, and $d$ is the hidden dimension of its feature vectors. After that, we apply a quantizer to convert this feature map $\mathbf{Z}$ into a discrete tokens map $\mathbf{Q} \in \mathbb{V}^{T' \times H' \times W'}$ (illustrated in Fig. 1), where $\mathbb{V}$ represents a visual vocabulary of size $|\mathbb{V}| = K$. After tokenization, these discrete tokens $\mathbf{Q}$ can be passed through a causal 3D CNN decoder to reconstruct the video $\hat{\mathbf{X}}$. We note that MAGVITv2 has two major advantages:

**(1) Joint Image-Video Tokenization.** MAGVITv2 allows to tokenize images and videos with a shared vocabulary. To achieve this, the number of frames in an input video, $T$, must satisfy $T = 1 + n \times F_T$, meaning the video comprises an initial frame followed by $n$ clips, each containing $F_T$ frames. When $n = 0$, the video contains only the initial frame, thus simplifying the video to an image. Both the initial frame and each subsequent clip are discretized into a $(1, H', W')$ token map. Therefore, the latent temporal dimension $T'$ of the token map $\mathbf{Q}$ equals to $1 + n$, which achieves $F_T$ times downsampling ratio on the temporal dimension (except for the first frame). Additionally, $H' = \frac{H}{F_H}$ and $W' = \frac{W}{F_W}$, where $F_H, F_W$ are spatial downsampling factors.

**(2) Temporal Causality.** The causal 3D CNN architecture ensures that the tokenization and detokenization of each clip depend only on the preceding clips, facilitating autoregressive modeling along the temporal dimension, which will be discussed further in § 3.3.

---

[2]Images can be considered as "static" videos with $T = 1$.

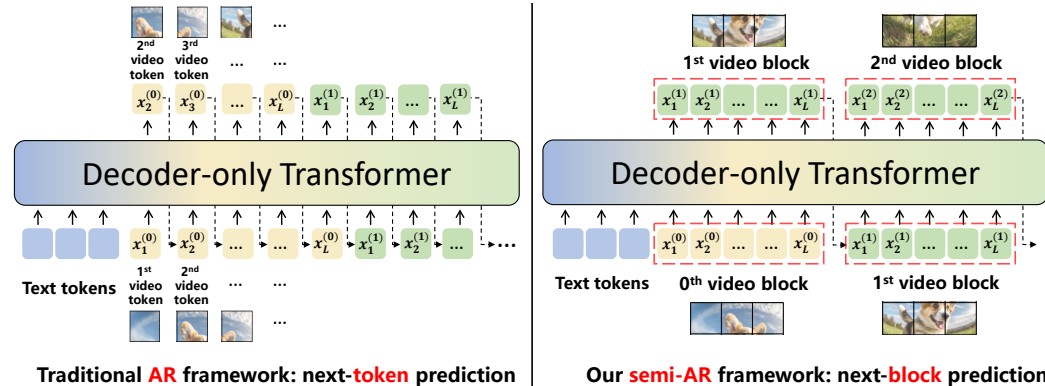

Figure 3: Comparison between a vanilla auto-regressive (AR) framework based on next-token prediction (left) and our semi-AR framework based on next-block prediction (right). $x_j^{(i)}$ indicates the $j^{th}$ video token in the $i^{th}$ block, with each block containing $L$ tokens. The dashed line in the right panel presents that the $L$ tokens generated in the current step are duplicated and concatenated with prefix tokens, forming the input for the next step's prediction during inference.

## 3.2 PRELIMINARY: AUTO-REGRESSIVE MODELING FOR VIDEO GENERATION

Inspired by the success of AR models in the field of NLP, previous work (Yan et al., 2021; Wu et al., 2021a;b) has extended AR models to video generation. Typically, these methods flatten the 3D video token input $\mathbf{Q} \in \mathbb{V}^{T' \times H' \times W'}$ into a 1D token sequence. Let $C^{(t)} = \{x_1^{(t)}, x_2^{(t)}, \dots, x_L^{(t)}\}$ be the set of tokens in the $t^{th}$ clip, where $L = H' \times W' = |C^{(t)}|$ is the total number of tokens in each clip, and every clip contains an equal number of tokens. Specially, when $t = 0$, $C^{(0)}$ denotes the first frame's tokens. Therefore, the 1D token sequence can be represented as ( $C^{(0)}$ , ..., $C^{(T')}$ ) = ( $x_1^{(0)}, x_2^{(0)}, \dots, x_L^{(0)}$ , ..., $x_1^{(T')}, x_2^{(T')}, \dots, x_L^{(T')}$ ). In the AR framework, the next-token probability is conditioned on the preceding tokens, where each token $x_l^{(t)}$ depends only on its prefix $(x_l^{(<t)}, x_{<l}^{(t)})$. This unidirectional dependency allows the likelihood of the 1D sequence to be factorized as:

$$p\left(x_1^{(0)}, \dots, x_L^{(T')}\right) = \prod_{t=1}^{T'} \prod_{l=1}^{L} p\left(x_l^{(t)} \mid x_l^{(<t)}, x_{<l}^{(t)}\right) \qquad (1)$$

Since only one token is predicted per step, the inference process can become computationally expensive and time-consuming, motivating the exploration of more efficient methods, such as semi-AR models, to improve both speed and scalability.

## 3.3 SEMI-AR MODELING VIA NEXT BLOCK MODELING

In contrast to text, which consists of variable-length words and phrases, video content can be uniformly decomposed into equal-sized blocks (e.g., rows or frames). Fig. 2 shows examples of token-wise, row-wise, and frame-wise block representations. Based on this, we propose a semi-autoregressive (semi-AR) framework named **N**ext-**B**lock **P**rediction (NBP), where each token in the current block predicts the corresponding token in the next block. Fig. 3 illustrates an example of next-clip prediction, where each clip is treated as a block, and the next clip is predicted based on the preceding clips. This approach introduces two key differences compared to vanilla NTP modeling:
**(1) Change in the generation target.** In NBP, the $l^{th}$ token $x_l^{(t)}$ in the $t^{th}$ clip predicts $x_l^{(t+1)}$ in the next clip, rather than $x_{l+1}^{(t)}$ as in NTP. **(2) Increase in the number of generation targets.** Instead of predicting one token at a time, all $L$ tokens $x_{1:L}^{(t)}$ simultaneously predict the corresponding $L$ tokens

Table 1: Video reconstruction results on UCF-101 and K600.

| Method | Backbone | Quantizer | Param. | # bits | UCF-101 | | | | K600 | | | |
|---|---|---|---|---|---|---|---|---|---|---|---|---|
| | | | | | rFVD↓ | PSNR↑ | SSIM↑ | LPIPS↓ | rFVD↓ | PSNR↑ | SSIM↑ | LPIPS↓ |
| MaskGIT Chang et al. (2022) | 2D CNN | VQ | 53M | 10 | 216 | 21.5 | .685 | .1140 | - | - | - | - |
| TATS Ge et al. (2022) | 3D CNN | VQ | 32M | 14 | 162 | - | - | - | - | - | - | - |
| OmniTokenizer Wang et al. (2024) | ViT | VQ | 78M | 13 | 42 | 30.3 | .910 | .0733 | 27 | 28.5 | .883 | .0945 |
| MAGVIT-v1 Yu et al. (2023b) | 3D CNN | VQ | 158M | 10 | 25 | 22.0 | .701 | .0990 | - | - | - | - |
| MAGVIT-v2 Yu et al. (2024) | C.-3D CNN | LFQ | 158M | 18 | 16.12 | - | - | .0694 | - | - | - | - |
| MAGVIT-v2 Yu et al. (2024) | C.-3D CNN | LFQ | 370M | 18 | 8.62 | - | - | .0537 | - | - | - | - |
| Ours | C.-3D CNN | FSQ | 370M | 16 | 15.50 | 29.3 | .893 | .0648 | 6.73 | 31.3 | .944 | .0828 |

$x_{1:L}^{(t+1)}$ in the next clip. Accordingly, the NBP objective function can be expressed as:

$$p\left(x_1^{(0)}, \ldots, x_L^{(T')}\right) = \prod_{t=1}^{T'} p\left(x_{1:L}^{(t)} \mid x_{1:L}^{(0)}, \ldots, x_{1:L}^{(t-1)}\right) \tag{2}$$

By adjusting the block size, the framework can generate videos using different generation units. To ensure the effectiveness of this approach, three key components are designed:

**(1) Initial Condition.** In NTP models, a special token (e.g., [begin_of_video]) is typically used as the initial condition. In the NBP setting, we can add a block of special tokens to serve as the initial condition for generating the first block. However, to simplify learning and enhance control over the generated video, we use the first frame $C^{(0)}$ as the initial condition. In practice, following Girdhar et al. (2023), users can upload an image as the first frame, or call a off-the-shelf text-to-image model (e.g., SDXL (Podell et al., 2023)) to generate it. Besides, both NTP and NBP models can accept various inputs (e.g., text) as conditions (see Fig. 3).

**(2) Block-wise Attention.** To better capture spatial dependency, we allows tokens to attend to all tokens within the same block via bidirectional attention. Fig. 4 compares traditional causal attention in NTP modeling with block-wise attention in NBP modeling.

**(3) Inference Process.** To illustrate the inference process of next-block prediction, we consider a scenario where each block corresponds to a clip. As shown in the right panel of Fig. 3, during inference, the last $L$ tokens of the current output represents the predicted tokens for the next block. These tokens are retained and concatenated with clip prefix, forming the input for

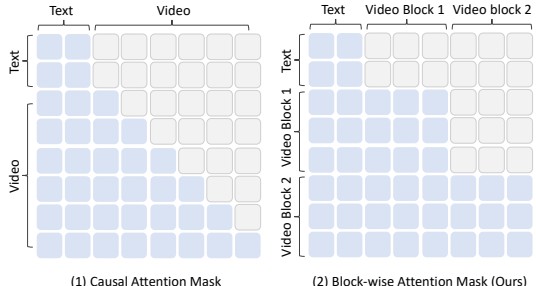

Figure 4: Causal attention mask in NTP modeling v.s. block-wise attention mask in NBP modeling.

the next step. By transitioning from token-by-token to block-by-block prediction, the NBP framework leverages parallelization, reducing the number of generation steps by a factor of $L$, thereby decreasing computational cost and accelerating inference.

## 4 EXPERIMENTS

### 4.1 EXPERIMENTAL SETUPS

**Video Tokenizer.** In contrast to the official implementation of MAGVITv2, which utilizes LFQ (Yu et al., 2024) as its quantizer, we adopt FSQ (Mentzer et al., 2023) due to its simplicity and reduced number of loss functions and hyper-parameters. Following the original paper's recommendations, we set the FSQ levels to $[8, 8, 8, 5, 5, 5]$, and the size of the visual vocabulary is 64K. Moreover, we employ PatchGAN (Isola et al., 2016) instead of StyleGAN (Karras et al., 2018) to enhance training stability. The reconstruction performance of our tokenizer is presented in Table 1, and additional training details are available in the Appendix A.2. We note that MAGVITv2 is not open-sourced, we have made every effort to replicate its results. Our tokenizer surpasses OmniTokenizer Wang et al.

Table 2: Comparison of next-token prediction (NTP) and next-block prediction (NBP) models in terms of performance and speed, evaluated on the K600 dataset (5-frame condition, 12 frames (768 tokens) to predict). Inference time was measured on a single A100 Nvidia GPU. All models are implemented by us under the same setting and trained for 20 epochs. FPS denote "frame per second".

| Model Size | Modeling Method | # Block size | FVD ↓ | # Forward steps | Inference speed (FPS) ↑ |
|---|---|---|---|---|---|
| 700M | NTP | 1 (1×1×1) | 38.5 | 768 | 0.80 |
| | NBP (Ours) | 16 (1×1×16) | **33.6** | **48** | **8.89** |
| 1.2B | NTP | 1 (1×1×1) | 32.2 | 768 | 0.75 |
| | NBP (Ours) | 16 (1×1×16) | **28.6** | **48** | **6.70** |
| 3B | NTP | 1 (1×1×1) | 28.1 | 768 | 0.60 |
| | NBP (Ours) | 16 (1×1×16) | **26.5** | **48** | **4.29** |

(2024), MAGVITv1 Yu et al. (2023b), and other models in performance. However, due to limited computational resources, we did not pre-train on ImageNet (Russakovsky et al., 2014) or employ a larger visual vocabulary (e.g., 262K as in the original MAGVITv2), which slightly impacts our results compared to the official MAGVITv2. Nevertheless, we note that the primary objective of this paper is to validate the semi-AR framework, rather than to achieve state-of-the-art tokenizer performance.

**Generator Training Details.** We train decoder-only transformers on 17-frame videos with a resolution of $128 \times 128$, using the UCF-101 (Soomro et al., 2012) and K600 (Carreira et al., 2018) datasets. With spatial downsampling factors of $F_H = F_W = 8$ and temporal downsampling of $F_T = 4$, the resulting 3D token map for each video sample has dimensions $(T', H', W') = (5, 16, 16)$, yielding a total of 1280 tokens. We train our model for 100K steps with a total batch sizes of 256 and 64 respectively. Model sizes range from 700M to 3B parameters, with training spanning approximately two weeks on 32 NVIDIA A100 GPUs. The full model configuration and training hyper-parameters are provided in Appendix A.2. We train the models from scratch, rather than initializing from a pre-trained LLM checkpoint, as these text-based checkpoints provide minimal benefit for video generation (Zhang et al., 2023). We use LLaMA (Touvron et al., 2023) vocabulary (32K tokens) as the text vocabulary and merge it with the video vocabulary (64K tokens) to form the final vocabulary. Since our primary focus is video generation, we compute the loss only on video tokens, which leads to improved performance.

**Evaluation protocol.** We evaluate our models on UCF-101 and K600 datasets. Standard metrics such as Fréchet Video Distance (FVD) Unterthiner et al. (2018) are used to assess video quality, while frame-level metrics including PSNR, SSIM Wang et al. (2004) and LPIPS Zhang et al. (2018) are also reported. Additional evaluation details are provided in Appendix A.4.

## 4.2 COMPARISON OF NEXT-TOKEN PREDICTION AND NEXT-BLOCK PREDICTION

We first conduct a fair comparison between next-token prediction (NTP) and our next-block prediction (NBP) under the same experimental setting. Table 2 highlights the superiority of our approach in three key aspects: generation quality, inference efficiency, and scalability.

**Generation Quality.** Across all model sizes, NBP with a $1 \times 1 \times 16$ block size consistently outperforms NTP models in terms of generation quality (measured by FVD). For instance, the 700M NBP model achieves an FVD of 33.6, outperforming the NTP model by 4.9 points. Furthermore, a NBP model with only 1.2B parameters achieves a comparable performance to a 3B NTP model (28.6 vs. 28.1 FVD). This sug-

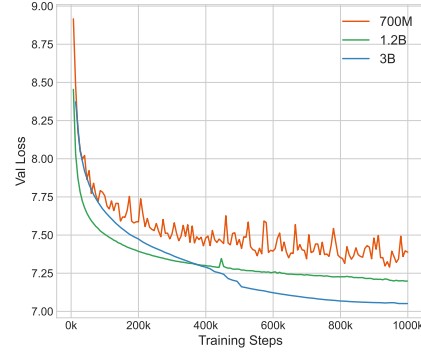

Figure 5: Validation loss of various size of semi-AR models from 700M to 3B.

Table 3: Comparions of class-conditional generation results on UCF-101 and frame prediction results on K600. MTM indicates mask token modeling. Our model on K600 is trained for 77 epochs, we gray out models that use significantly more training computation (e.g., those trained for over 300 epochs) for a fair comparison.

| Type | Method | #Param | UCF-101 | | | | K600 | | | |
|------|--------|--------|------|------|------|------|------|------|------|------|
| | | | FVD↓ | PSNR↑ | SSIM↑ | LPIPS↓ | FVD↓ | PSNR↑ | SSIM↑ | LPIPS↓ |
| GAN | DVD-GAN (Clark et al., 2019) | N/A | - | - | - | - | 31.1 | - | - | - |
| Diffusion | VideoFusion (Luo et al., 2023) | N/A | 173 | - | - | - | - | - | - | - |
| Diffusion | Make-A-Video (Singer et al., 2022) | N/A | 81.3 | - | - | - | - | - | - | - |
| Diffusion | HPDM-L (Skorokhodov et al., 2024) | 725M | 66.3 | - | - | - | - | - | - | - |
| MTM | Phenaki Villegas et al. (2022) | 227M | - | - | - | - | 36.4 | - | - | - |
| MTM | MAGVIT Yu et al. (2023b) | 306M | 76 | - | - | - | 9.9 | - | - | - |
| MTM | MAGVITv2 Yu et al. (2024) | 840M | 58 | - | - | - | 4.3 | - | - | - |
| AR | LVT Rakhimov et al. (2020) | 50M | - | - | - | - | 224.7 | - | - | - |
| AR | ViTrans Weissenborn et al. (2020) | 373M | - | - | - | - | 170.0 | - | - | - |
| AR | CogVideo Hong et al. (2023) | 9.4B | 626 | - | - | - | 109.2 | - | - | - |
| AR | ViVQVAE Walker et al. (2021) | N/A | - | - | - | - | 64.3 | - | - | - |
| AR | TATS Ge et al. (2022) | 321M | 332 | - | - | - | - | - | - | - |
| AR | OmniTokenizer Wang et al. (2024) | 227M | 314 | - | - | - | 34.2 | - | - | - |
| AR | OmniTokenizer Wang et al. (2024) | 650M | 191 | - | - | - | 32.9 | 21.4 | .781 | .061 |
| Semi-AR | NBP (Ours) | 700M | 55.0 | 22.6 | .708 | .115 | 25.5 | 21.1 | .724 | .070 |
| Semi-AR | NBP (Ours) | 1.2B | 34.0 | 23.4 | **.749** | .113 | 23.0 | **21.2** | .727 | .069 |
| Semi-AR | NBP (Ours) | 3B | **20.7** | **24.6** | .749 | **.109** | **19.5** | **21.2** | **.728** | **.068** |

gests that the block size of $1\times1\times16$ is a more effective generation unit for auto-regressive modeling in video domain.

**Inference Efficiency.** For generating a 12-frame video ($128\times128$ resolution, 768 tokens), a 700M NTP model requires 768 forward step during inference, taking 15.04 seconds (FPS=0.80). In contrast, our NBP model with a $1\times1\times16$ block size predicts all tokens in a row simultaneously, requiring only 48 steps and 1.35 seconds to generate the video (FPS=8.89)—over 11 times faster than the NTP model. Since NBP modifies only the target output and attention mask, it is compatible with most efficient AR inference frameworks, such as Flash Attention (Dao et al., 2022), offering potential for further speed improvements.

**Scalability.** As model size increases from 700M to 1.2B and 3B parameters, the FVD of NBP models improves from 33.6 to 28.6 and 26.5, respectively. This demonstrates that NBP exhibits similar scalability to NTP models, with the potential for even greater performance as model size and computational resources increase. Fig. 5 and Fig. 14 present the validation loss curves and generation examples for different model sizes, respectively. As the models grow larger, the generated content exhibits greater stability and enhanced visual detail.

## 4.3 BENCHMARKING WITH PREVIOUS SYSTEMS

Table 3 presents our model's performance compared to strong baselines using various modeling approaches, including GAN, diffusion, mask token modeling (MTM), and vanilla auto-regressive (AR) methods. For UCF-101, the evaluation task is class-conditional video generation, where models generate videos based on a given class name. Since our method utilizes an image as initial visual condition, alongside the classname, we take the first frame from the training videos into condition additionally. This ensures no information leakage from the test set. Our Semi-AR model, with 700M parameters, achieves an FVD of 55.0, surpassing HPDM-L (Skorokhodov et al., 2024) and MAGVITv2 Yu et al. (2024) by 11.3 and 3 FVD points, respectively.

For K600, the evaluation task is frame prediction, where all models predict future frames based on the same 5-frame condition from the validation set. Our 700M model achieves an FVD of 25.5, outperforming the strongest AR baseline, OmniTokenizer, by 7.4 FVD points. While our model exhibits a performance gap compared to MAGVITv2, it achieves this result with significantly lower

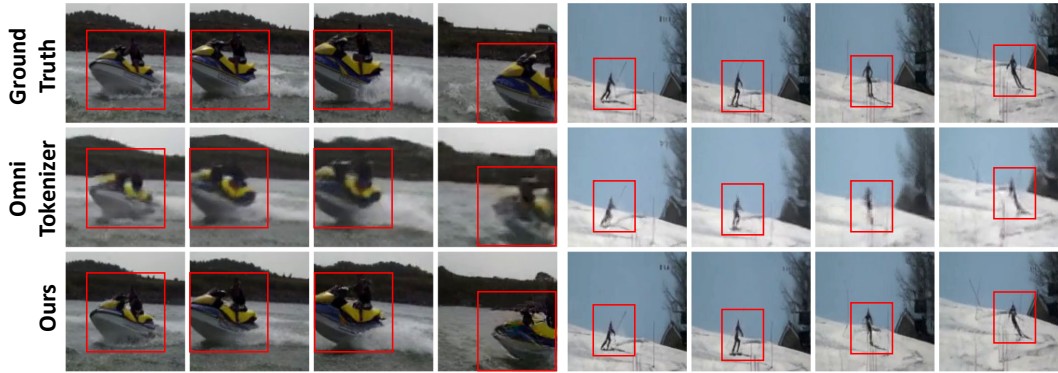

Figure 6: Video reconstruction results (17 frames 128×128 resolution at 25 fps and shown at 6.25 fps) of OmniTokenizer and our method.

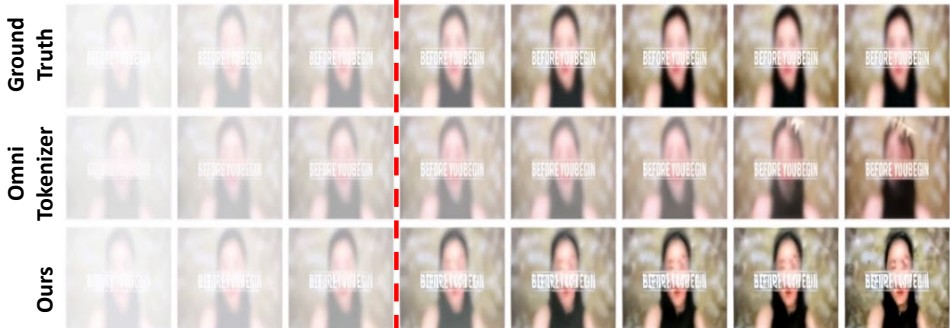

Figure 7: Frame prediction results of OmniTokenizer and our method. The left part is the condition, the right part is predicted subsequent sequence.

training computation (e.g., 77 epochs vs. MAGVITv2's 360 epochs). Scaling up the model size narrows this gap, with a 6-point improvement in FVD observed. Given the strong scalability of our semi-AR framework, we believe that with larger model sizes and increased training volumes, our approach could surpass MAGVIT2, akin to how large language models (LLMs) (Brown et al., 2020) have outperformed BERT (Devlin, 2018) in NLP.

### 4.4 VISUALIZATIONS

**Video Reconstruction.** Fig. 6 compares the video reconstruction results of OmniTokenizer (Wang et al., 2024) and our tokenizer. Our method significantly outperforms the baseline in both image clarity and motion stability.

**Video Generation.** Fig. 7 and 10 showcase the frame prediction results generated by our model. The visualizations demonstrate that our model accurately predicts subsequent frames with high clarity and temporal coherence, even in scenarios involving large motion dynamics. Fig. 13 shows more generation results of our 3B model.

### 4.5 ABLATION STUDY AND ANALYSIS

In this subsection, we conduct an ablation study on block size and analyze the attention patterns in our NBP models.

**Ablation Study on Block Size.** We experiment with different block sizes, ranging from $[1, 16, 64, 256]$ [3], to assess their impact on model performance. A block size of 1, 16, and 256

---

[3]The full 3D size of the blocks are $1{\times}1{\times}1$, $1{\times}1{\times}16$, $1{\times}4{\times}16$, $1{\times}16{\times}16$, respectively.

corresponds to token-by-token (NTP), row-by-row, and clip-by-clip generation, respectively. Fig. 8 demonstrates the training loss curves for various block sizes. As block size decreases, learning becomes easier due to the increased prefix conditioning, which simplifies the prediction task. However, using the smallest block size (i.e., a single token) does not yield optimal performance. As shown in Fig. 9, a block size of 16 achieves the best generation quality, with an FVD improvement of 3.5 points, reaching 25.5. Block size plays a critical role in balancing generation quality (FVD) and efficiency (FPS). While larger blocks (e.g., $1\times16\times16$) result in faster inference speeds (up to 17.14 FPS), performance degrades, suggesting that generating an entire clip in one step is overly challenging. Additionally, inference decoding methods significantly influence results. As demonstrated in Fig. 15, traditional Top-P Top-K decoding can lead to screen fluctuations, as it struggles to model spatial dependencies within large blocks, highlighting the need for improved decoding strategies in NBP scenarios.

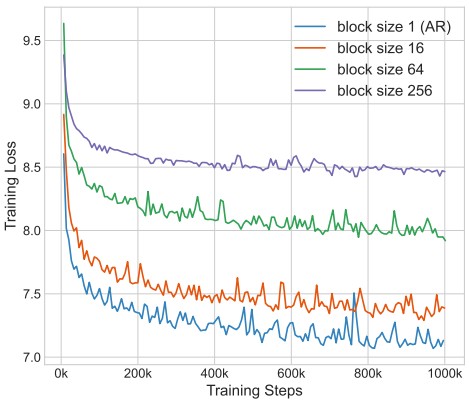

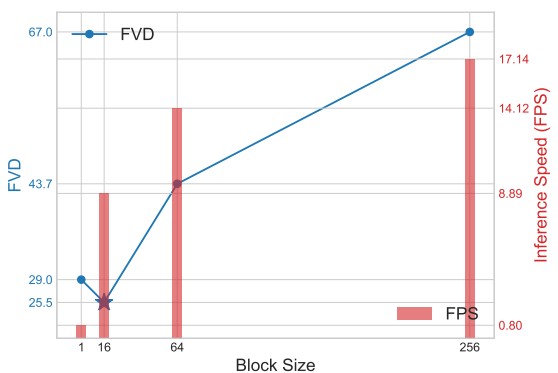

Figure 8: Training loss of various block sizes from 1 to 256.

Figure 9: Generation quality (FVD, lower is better) and inference speed (fps, higher is better) of various block sizes from 1 to 256.

**Analysis of Attention Pattern.** We analyze the attention pattern in our NBP framework using an example of next-clip prediction, where each block corresponds to a clip. Fig. 11 shows the attention weights on UCF-101. Unlike the lower triangular distribution observed in AR models, our attention is characterized by a staircase pattern across blocks. In addition to high attention scores along the diagonal, the map reveals vertical stripe-like highlighted patterns, indicating that tokens at certain positions receive attention from all tokens. Fig. 12 illustrates the spatial attention distribution for a specific query (marked by red ×). This query can attend to all tokens within the clip, rather than being restricted to only the preceding tokens in a raster-scan order, enabling more effective spatial dependency modeling.

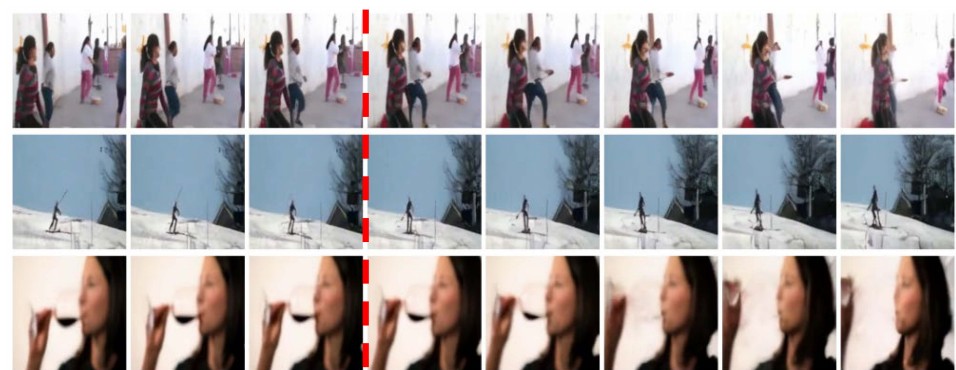

Figure 10: Visualization of frame prediction results of our method.

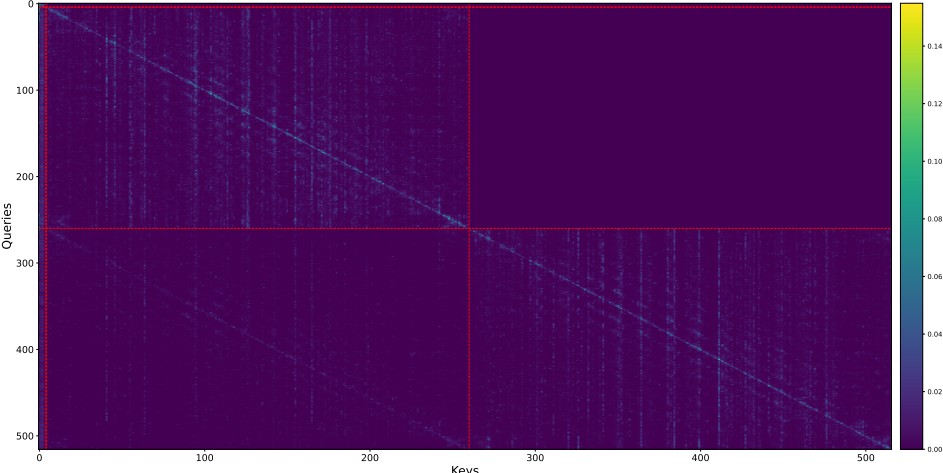

Figure 11: Attention weights of next-clip prediction on UCF-101. The horizontal and vertical axis represent the keys and queries, respectively. Two red lines on each axis divide the axis into three segments, corresponding to the text (classname), the first clip, and the second clip. The brightness of each pixel reflects the attention score. We downweight the attention to text tokens by $5\times$ to provide a more clear visualization.

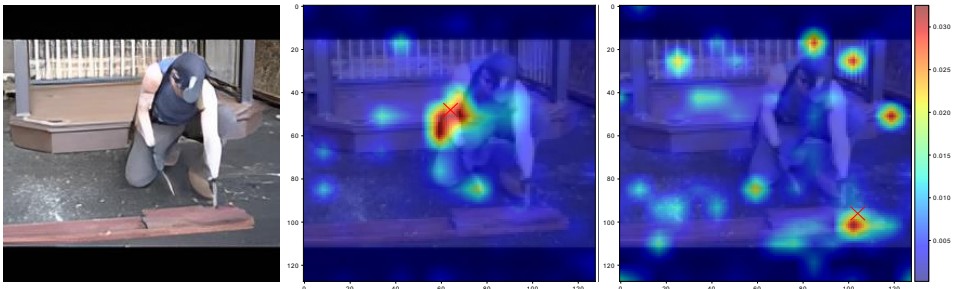

Figure 12: Spatial attention distribution for a specific query (represented by red $\times$) on UCF-101.

## 5 CONCLUSION

In this paper, we introduced a novel approach to video generation called Next Block Prediction using a semi-auto-regressive modeling framework. This framework offers a more efficient and scalable solution for video generation, combining the advantages of parallelization with improved spatial-temporal dependency modeling. This method not only accelerates inference but also maintains or improves the quality of generated content, demonstrating strong potential for future applications in multimodal AI.

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

Table 4: Model sizes and architecture configurations of our generation model. The configurations are following LLaMA (Touvron et al., 2023).

| Model | Parameters | Layers | Hidden Size | Heads |
|---|---|---|---|---|
| NBP-XL | 700M | 24 | 1536 | 16 |
| NBP-XXL | 1.2B | 24 | 2048 | 32 |
| NBP-3B | 3B | 32 | 3072 | 32 |

## A  IMPLEMENTATION DETAILS

### A.1  TASK DEFINITIONS

We introduce the tasks used in our training and evaluation. Each task is characterized by a few adjustable settings such as interior condition shape and optionally prefix condition. Given a video of shape $T \times H \times W$, we define the tasks as following:

- Class-conditional Generation (CG)
    - Prefix condition: class label.
- Class-conditional Frame Prediction (CFP)
    - Prefix condition: class label.
    - Interior condition: $t$ frames at the beginning; $t = 1$.
- Frame Prediction (FP)
    - Interior condition: $t$ frames at the beginning; $t = 5$ for K600 dataset.

As we stated in § 4.3, for UCF-101, other baselines perform the CG task, while our models perform the CFP task, as our method utilizes an image as initial visual condition, alongside the classname. We take the first frame from the training videos into condition additionly. This ensures no information leakage from the test set. For K600, all the methods perform the FP task.

### A.2  MODEL CONFIGURATION

**Video Tokenizer.**  Our video tokenizer shares the same model architecture with MAGVITv2 Yu et al. (2024).

**Decoder-only Generator.**  Table 4 shows the configuration for decoder-only generator. We use separate position encoding for text and video. We do not use advanced techniques in large language models, such as rotary position embedding (RoPE) (Su et al., 2024), SwiGLU MLP, or RMS Norm (Touvron et al., 2023), which we believe could bring better performance.

### A.3  TRAINING

**Video Tokenizer.**  Table 5 shows training configurations of our video tokenizer.

**Decoder-only Generator.**  Table 6 shows training configurations of our video generator.

For both tokenizer and generator training, the video samples are all 17 frames, frame stride 1, 128×128 resolution.

### A.4  EVALUATION

**Evaluation metrics.**  The FVD Unterthiner et al. (2018) is used as the primary evaluation metric. We follow the official implementation[4] in extracting video features with an I3D model trained

---

[4]`https://github.com/google-research/google-research/tree/master/frechet_video_distance`

Table 5: Training configurations of video tokenizer.

| Hyper-parameters | UCF101 | K600 |
|---|---|---|
| Video FPS | 8 | 8 |
| Latent shape | 5×16×16 | 5×16×16 |
| Vocabulary size | 64K | 64K |
| Embedding dimension | 6 | 6 |
| Initialization | Random | Random |
| Peak learning rate | 5e-5 | 1e-4 |
| Learning rate schedule | linear | linear |
| Warmup ratio | 0.01 | 0.01 |
| Perceptual loss weight | 0.1 | 0.1 |
| Generator adversarial loss weight | 0.1 | 0.1 |
| Optimizer | Adam | Adam |
| Batch size | 256 | 256 |
| Epoch | 2000 | 100 |

on Kinetics-400 Carreira & Zisserman (2017). We further include image quality metrics: PSNR, SSIM Wang et al. (2004) and LPIPS Zhang et al. (2018) (computed by the VGG features).

**Sampling protocols.** We follow the sampling protocols from previous works Yu et al. (2024); Ge et al. (2022); Clark et al. (2019) when eveluating on the standard benchmarks, i.e. UCF-101, and Kinetics-600. We sample 17-frame clips from each dataset without replacement to form the real distribution in FVD and extract condition inputs from them to feed to the model. We continuously run through all the samples required (e.g., 40,000 for UCF-101) with a single data loader and compute the mean and standard deviation for 4 folds.

Below are detailed setups for each dataset:

- UCF-101:
    - Dataset: 9.5K videos for training, 101 classes.
    - Number of samples: 10,000×4.
    - Resolution: 128×128.
    - Real distribution: random clips from the training videos.
- Kinetics-600:
    - Dataset: 384K videos for training and 29K videos for evaluation.
    - Number of samples: 50,000×4.
    - Generation resolution: 128×128.
    - Evaluation resolution: 64×64, via central crop and bilinear resize.

# B VISUALIZATION

We provide additional visualization of video generation results. Fig. 13 shows results of our 3B model. Fig. 14 shows results of various model size (700M, 1.2B and 3B). Fig. 15 shows results of various block size (1×1×1, 1×1×16 and 1×16×16).

Table 6: Training configurations of video generator (base model).

| Hyper-parameters | UCF101 | K600 |
|---|---|---|
| Video FPS | 8 | 16 |
| Latent shape | 5×16×16 | 5×16×16 |
| Vocabulary size | 96K (including 32K text tokens) | 64K |
| Initialization | Random | Random |
| Peak learning rate | 6e-4 | 1e-3 |
| Learning rate schedule | linear | linear |
| Warmup steps | 5,000 | 10,000 |
| Weight decay | 0.01 | 0.01 |
| Optimizer | Adam (0.9, 0.98) | Adam (0.9, 0.98) |
| Batch size | 256 | 64 |
| Epoch | 2560 | 77 |

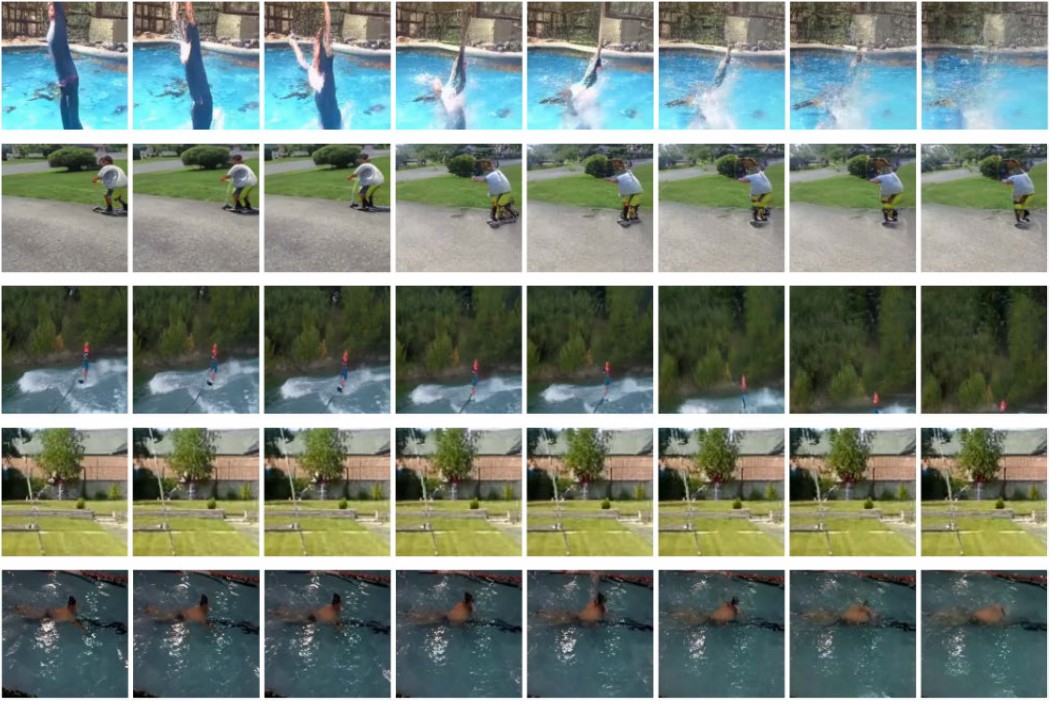

Figure 13: Visualization of video generation results of our 3B model.

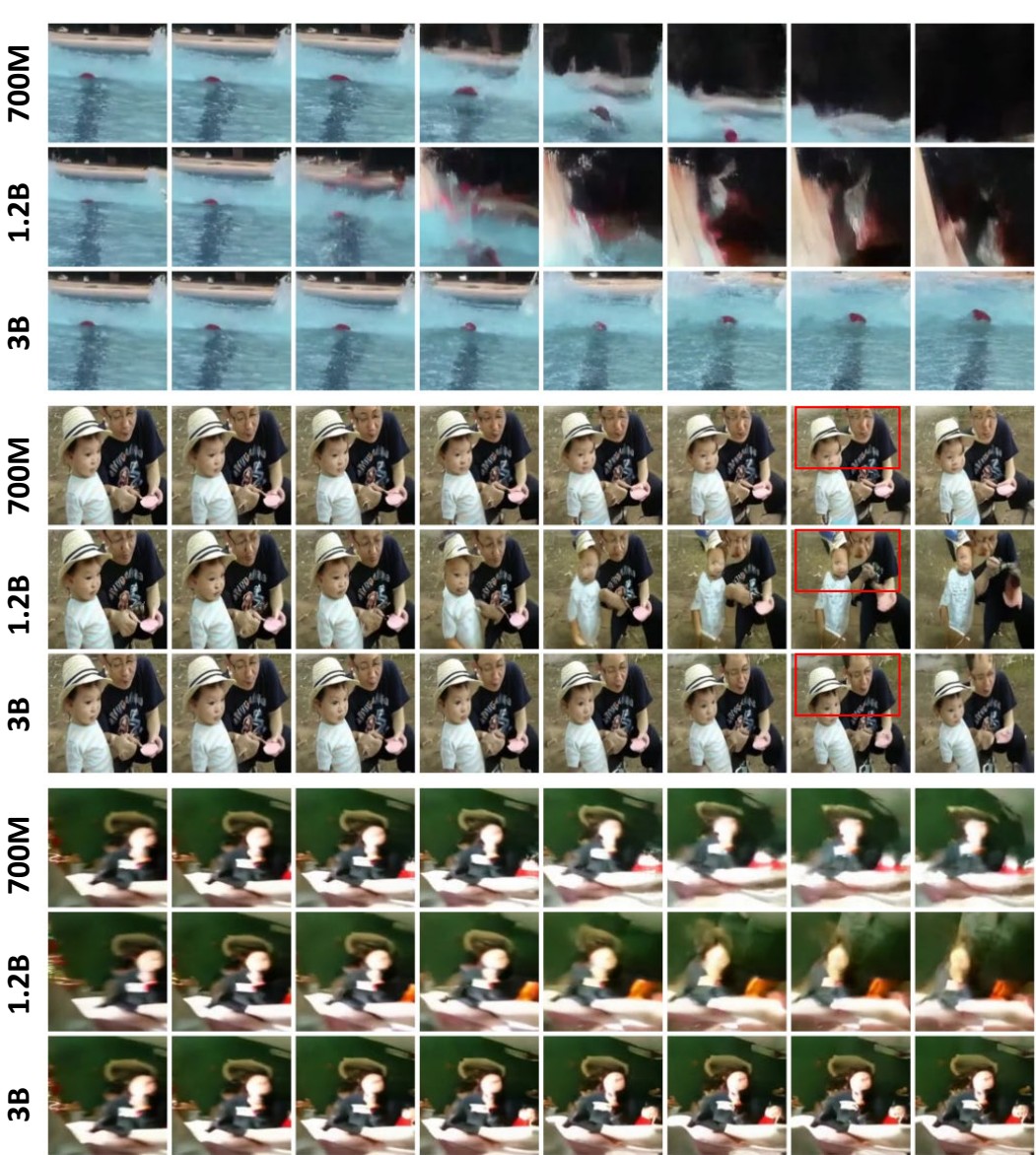

Figure 14: Visualization of video generation results of various model size (700M, 1.2B and 3B).

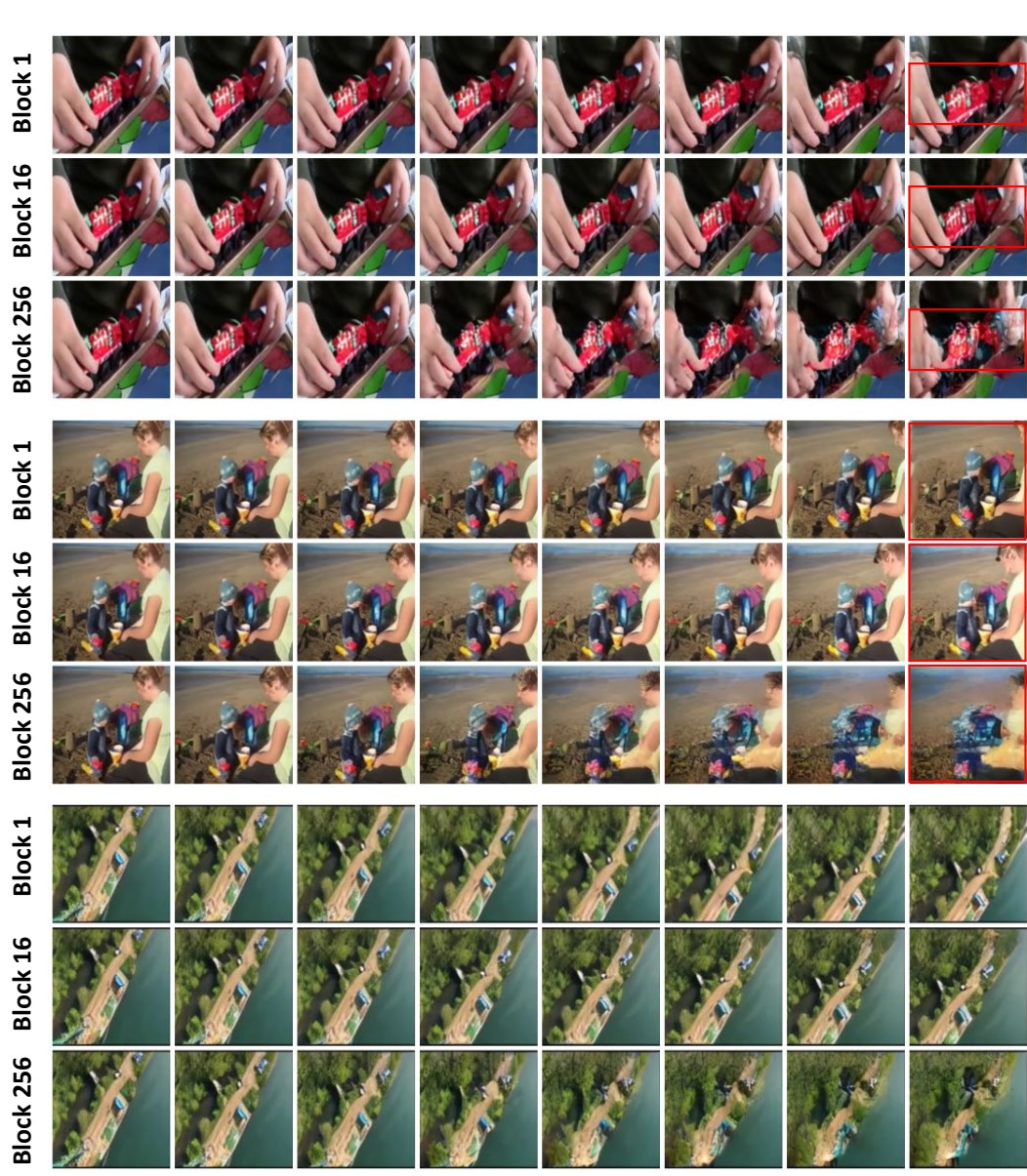

Figure 15: Visualization of video generation results of various block size ($1\times1\times1$, $1\times1\times16$ and $1\times16\times16$).

