# OpenReview forum: "Next Block Prediction: Video Generation via Semi-Auto-Regressive Modeling"
_ICLR.cc/2025/Conference — Submitted to ICLR 2025_

### Official Review · Reviewer_EmCQ · 2024-10-30

**Soundness:** 3
**Presentation:** 3
**Contribution:** 2
**Rating:** 5
**Confidence:** 4

**Summary:**

This paper introduces a semi-autoregressive video generation model that enables next-block prediction (NBP),  i.e., predicting multiple tokens in parallel. The NBP model uses a block-wise causal attention matrix, i.e., causal attention inter-block and bidirectional attention intra-block, capturing better spatial dependencies. Extensive experiments show the state-of-the-art video generation quality of NBP and a significant improvement in inference speed.

**Strengths:**

- This paper proposes a semi-autoregressive paradigm (i.e., next block prediction) for video generation, which brings better spatial dependencies in the attention computation and a significant inference speed improvement
- This paper provides extensive experiments in terms of the design choice for block division. The model shows a good trade-off between inference speed and generation quality.
- The writing and presentation of this paper are clear and easy-to-follow

**Weaknesses:**

- The technical innovation from "next token prediction" to "next block prediction" is a bit trivial.
  - Since there have been many studies on the semi-autoregressive paradigm (blockwise attention and parallel decoding) in the NLP [1,2] and vision[3,4]  fields, the work done in this paper is more like an engineering application rather than a technological innovation.
  - In addition to simply changing the model prediction and the attention map, this paper does not outline the technical challenges or insights encountered in modifying an AR model to semi-AR (i.e., from "next token prediction" to "next block prediction" ). The author(s) may provide some clarifications and insights in the rebuttal.

$\quad$
- The semi-AR (i.e., next-block-prediction) paradigm proposed in this paper does not seem to be restricted to video generation.  This means when an image tokenizer is used, it can also be applied to image generation. As a general semi-AR paradigm, quantitative comparisons on the  ImageNet dataset are suggested for drawing more convincing conclusions in this paper.

$\quad$
- The temporal axis is not considered in the block division for all design choices (e.g., 1x1x16, 1x4x16, and1x16x16) presented in this paper.
  - Is this because the video tokenizer (currently MAGVITv2 is used in the paper with a 4 x temporal downsampling) ?
  - How does dividing blocks along the temporal axis influence the results of video generation?  Further ablation studies are suggested.


$\quad$

[1] Stern, Mitchell, Noam Shazeer, and Jakob Uszkoreit. "Blockwise parallel decoding for deep autoregressive models." *Advances in Neural Information Processing Systems* 31 (2018).

[2] Leviathan, Yaniv, Matan Kalman, and Yossi Matias. "Fast inference from transformers via speculative decoding." *International Conference on Machine Learning*. PMLR, 2023.

[3] Li, Jiacheng, et al. "Lformer: Text-to-Image Generation with L-shape Block Parallel Decoding." *arXiv preprint arXiv:2303.03800* (2023).

[4] Tian, Keyu, et al. "Visual autoregressive modeling: Scalable image generation via next-scale prediction." *arXiv preprint arXiv:2404.02905* (2024).

**Questions:**

- In Figure 4,  when the first frame is added as the initial condition, the attention map should have extra columns right next to the text column. Should Figure 4 be adjusted, or is my understanding mistaken ?
- Is there any performance improvement in terms of the temporal consistency of generated videos when using NBP over NTP ?

---

> ### Author Response · Authors · 2024-11-26
> **Rebuttal by Authors (1)**
>
> Thank you for the valuable comments.
>
> > W1: The technical innovation from "next token prediction (NTP)" to "next block prediction (NBP)" is a bit trivial. There have been studies on the semi-AR paradigm (blockwise attention and parallel decoding) in the NLP and vision. This paper does not outline the technical challenges or insights encountered in modifying an AR model to semi-AR.
>
> R1:  **Our NBP framework, while straightforward, is far from trivial**. Transitioning from NTP to NBP in the context of video generation prompted us to rigorously investigate several critical questions: **(1) What is the most effective unit of generation in AR models— a single token or a block of multiple tokens?** If the latter, what size should this block be? **(2) Should AR modeling strictly employ causal mechanisms, or should it integrate both causal and bidirectional approaches?** If both, what should be the optimal mix and proportion?
>
> In response to these issues, we introduced the NBP framework. Our approach adapts the AR generation unit from a single token to a block, incorporating bidirectional attention within blocks and maintaining causal attention between them.
>
> As outlined in Section 2, **our approach diverges from existing techniques**. **(1)** Unlike speculative and parallel decoding in NLP, which rely on 1-to-N generation requiring multiple output heads, our N-to-N generation approach simplifies the process without necessitating additional output modules. **(2)** In comparison to vision approaches like Lformer and VAR, which employ L-shape and multi-scale decompositions respectively, our block decomposition strategy avoids the complexities of additional padding tokens or multi-scale tokenizers, aligning more closely with the GPT-style simplicity.
>
> **The shift from NTP to NBP involved substantial technical challenges, particularly in determining the optimal block size for generation**. As discussed in Section 4.5, smaller blocks (e.g., 1x1x1) were found inadequate due to limited spatial modeling capabilities and inefficiencies in inference, whereas larger blocks (e.g., 1×16×16) compromised performance by overcomplicating the generation task. Our experiments also highlighted the impact of decoding strategies on model performance. For instance, as shown in Fig.15, traditional Top-P Top-K decoding methods led to noticeable screen fluctuations when applied to large blocks, underscoring the necessity for refined decoding strategies in NBP scenarios.
>
> In conclusion, our investigations led us to adopt a 1x1x16 block configuration, which not only improved performance (demonstrated by a reduction of 4.4 in FVD on the K600 dataset) but also accelerated inference speeds by a factor of 11 compared to traditional NTP. The NBP framework not only retains the scalability of conventional AR models but also offers enhanced flexibility and simplicity for broader application across different modalities and tasks.
>
>
> > W2: As a general semi-AR paradigm, quantitative comparisons on the ImageNet dataset are suggested for drawing more convincing conclusions in this paper.
>
> R2: As discussed in Section 3.3 (1) on Initial Conditions, our framework is designed to start with an existing initial frame for generating subsequent blocks, making it unsuitable for image generation where no analogous "first frame" exists.
> We attempted to adapt the framework by introducing a special token block (e.g., [`begin_of_img`]) as an initial condition (thus supporting the generating of images or videos from scratch). However, this approach struggled with transitioning from the special tokens to generating meaningful content, resulting in suboptimal performance. Consequently, we did not pursue this method in our current work.
>
> In response to this issue, we are considering a hybrid method that could initially focus on next-token prediction and then transition to next-block prediction. We leave it for future work.

---

> ### Author Response · Authors · 2024-11-26
> **Rebuttal by Authors (2)**
>
> > W3: The temporal axis is not considered in the block division for all design choices.
>
> R3: We investigated different block shapes along the temporal axis:
> | Block Size | Block Shape (TxHxW) | FVD  |
> |:----------:|:-----------:|:----:|
> | 16         | 1x1x16      | **25.5** |
> | 16         | 2x1x8       | 29.2 |
> | 8          | 1x1x8       | **25.7** |
> | 8          | 2x2x2       | 32.7 |
>
> The results show that block shapes with T=1 generally perform better. This finding supports our initial design decision not to use blocks spanning multiple frames for two main reasons:
>
> **(1) Temporal Compression**: Given that input videos are sampled at 8 fps/16 fps and undergo 4x temporal downsampling during tokenization, there is already a high level of information compression in the temporal dimension. Modeling rapidly changing content simultaneously across frames proves challenging.
>
> **(2) Causality in Temporal Modeling**: Our goal is for the NBP framework not only to excel in video generation but also to serve as a potential world model. Since video represents the world in spatiotemporal dimensions and temporal changes in the world are causal, we aim to preserve complete causality in the temporal dimension during video generation. Block shape with T=1 avoids introducing bidirectional temporal attention, aligning with our philosophy of using an auto-regressive generator (a decoder-only transformer) and a tokenizer like MagVITv2 with T=1 as the temporal unit.
>
> In conclusion, our choice of using blocks with T=1 is strategic, aiming to effectively model the world's spatiotemporal dynamics while maintaining the causality of temporal changes.
>
> > Q1: In Figure 4, when the first frame is added as the initial condition, the attention map should have extra columns right next to the text column.
>
> A1: No, our approach employs block-wise attention for the first frame, consistent with subsequent frames, rather than full attention. In training, this block-wise attention allows the tokens in the first frame to contribute to the loss calculation. To ensure consistency, we maintain the same block-wise attention pattern during inference, even when the first frame is used as the initial condition.
>
> > Q2: Is there any performance improvement in terms of the temporal consistency of generated videos when using NBP over NTP?
>
> A2: Yes. We use VBench[1] suit to evaluate the subject consistency and background consistency of NTP and NBP. The results are:
> | Method | Block Shape | Subject Consistency | Background Consistency |
> |:------:|:-----------:|:-------------------:|:----------------------:|
> | NTP    | 1x1x1       |   91.42%  |  95.88%  |
> | NBP    | 1x1x16      |  92.54%  |  96.20%  |
>
> Our NBP framework shows better subject consistency and background consistency compared to NTP method.
>
> [1] https://github.com/Vchitect/VBench/tree/master

---

> > ### Comment · Reviewer_EmCQ · 2024-12-01
> >
> > I appreciate the authors' response and clarifications.
> >
> > My major concern with this paper is still the lack of core innovation.
> >
> > As mentioned in R1 and replied in W1, the authors take solving the two questions as their main contribution, i.e.,
> > >  (1) What is the most effective unit of generation in AR models— a single token or a block of multiple tokens? If the latter, what size should this block be? (2) Should AR modeling strictly employ causal mechanisms, or should it integrate both causal and bidirectional approaches?
> >
> > if so, the author should provide a thorough analysis of NBP and NTP, based on existing state-of-the-art AR models using NBP and NTP in both language and visual (image and video) generations. And the core innovation should be put on the  special considerations of NBP for video generation. This is also a consensus from reviewer gTpp.
> >
> > However, this paper failed to propose special designs for video. In other words, modifying an AR model in image generation with NBP's  block division and attention mask can do the same thing.
> >
> > In addition, the supplied experiments of different block shapes along the temporal axis show that $T=1$ is the best choice.  This further implies that this work does not have a strong correlation to **video** generation.
> >
> > As for the explanation:
> > > (2) Causality in Temporal Modeling: Our goal is for the NBP framework not only to excel in video generation but also to serve as a potential world model. Since video represents the world in spatiotemporal dimensions and temporal changes in the world are causal, we aim to preserve complete causality in the temporal dimension during video generation. Block shape with T=1 avoids introducing bidirectional temporal attention, aligning with our philosophy of using an auto-regressive generator (a decoder-only transformer) and a tokenizer like MagVITv2 with T=1 as the temporal unit.
> >
> > I can not agree with this explanation. If the authors claim that "we aim to preserve complete causality in the temporal dimension during video generation", it's completely contrary to the original spirit of NPB and semi-AR if claimed as a innovation to apply in video generation. Considering that NBP and semi-AR have already been successfully applied to NLP methods, this varifies that partial causal along the temporal axis (corresponds to the text sequence axis in NLP) should be successful.
> >
> > In summary, and after reading other reviewers' comments, I will keep my original score and lean toward rejection.

---

> > > ### Comment · Reviewer_gTpp · 2024-12-01
> > >
> > > I also checked the Reviewer EmCQ's comment, recognizing my comments are also mentioned.
> > > In overall, I also agreed with Reviewer EmCQ's comments.

---

> ### Author Response · Authors · 2024-12-01
> **Further response by authors**
>
> Dear reviewer EmCQ and gTpp, thanks for your response. We would like to re-clarify our special considerations of NBP for video generation, specifically our choice of T=1 for block design.
>
> > Considering that NBP and semi-AR have already been successfully applied to NLP methods, this varifies that partial causal along the temporal axis (corresponds to the text sequence axis in NLP) should be successful.
>
> We must respectfully disagree with this assertion. The application of NBP and semi-AR to NLP is NOT so successful. According to [1], applying NBP with a block length of 6 results in a 12% performance **degradation** in English-German translation tasks. In contrast, our implementation of NBP with a block length of 16 leads to a 4.4% **improvement** on the K600. This discrepancy highlights that applying NBP is not trivial since NBP inherently disrupts the sequential modeling nature that is crucial in NLP.
>
> To illustrate this point, consider the following generation sequence (`the original sprit of NBP and semi-AR if claimed`) with a block length of 3:
> | step 1 | | step2 | | step 3|
> |:--:|:--:|:--:|:--:|:--:|
> | the | -> | of | -> | semi-AR |
> | original | -> | NBP | -> | if |
> | spirit | -> | and | -> | claimed |
>
> As you can see, the generated sequence (e.g., `of NBP and` in step 2) appears unnatural and lacks coherence. The correspondence of `the -> of -> semi-AR` is also very strange. This issue is exacerbated when we attempt to apply a fixed block length along the temporal axis in video generation or the text sequence axis in NLP, where it is challenging to find a coherent, complete, and self-consistent "block." Such a rigid structure can significantly impair the generation quality.
>
> In contrast, when dealing with frames in videos, we can uniformly decompose them into equal-sized blocks with coherent semantics (e.g., rows), making it a much more natural and effective approach. This is the essence of our special considerations of NBP for video generation, and it is what leads to better performance in frame prediction tasks, such as those evaluated on the K600.
>
> [1] Wang et al., Semi-Autoregressive Neural Machine Translation, EMNLP2018.

---

> ### Comment · Reviewer_EmCQ · 2024-12-03
>
> Thank you for the clarifications about the $T=1$ case. I agree that $T>1$ is not the best case in NLP tasks as in [1].
>
> However, as stated in [1]:
> > It is not fair to directly compare BLEU and latency. A fairer way is to compare performance degradation and
> speedup, which are calculated based on their own baseline.
>
> In terms of the ablations for block sizes,  as a semi-AR work for video generation, the paper should analyze the impact of different $T$ on generation quality and speed, rather than simply showing that 1x1x16 block is faster than 1x1x1 block and has better FVD than 2x1x8 block, which is a bit trivial.
>
>
> In other words, the above concerns agree with one of the comments from Reviewer gTpp:
>
> >
> > 2. On the ablation study with various block size.
> >
> > Different from the claims/motivations in the paper, the ablation study's results show that the effectiveness of semi-AR for image/video generation should aim to learn non-correlated pixels/latents in parallel, instead of considering the spatial-temporal relationship between pixels/latents. Based on these results, the paper should be re-written to make it have more logical flow.
>
>
> In addition to the above points, I keep my comments about other weaknesses of this paper and keep my rating unchanged.
>
> [1] Wang et al., Semi-Autoregressive Neural Machine Translation, EMNLP2018.

---

### Official Review · Reviewer_G3HK · 2024-11-02

**Soundness:** 3
**Presentation:** 3
**Contribution:** 2
**Rating:** 3
**Confidence:** 4

**Summary:**

This paper introduces Next Block Prediction (NBP) for video generation, extending Next Token Prediction (NTP) to predict multiple tokens (a "block") at once. By shifting the prediction unit from a single token to a block, NBP achieves an 11x speedup and better performance than NTP during inference.

**Strengths:**

- The method is simple yet delivers better speed and performance than NTP.
- The scalability of the NBP framework is well-demonstrated in the paper.
- The writing is clear and easy to follow.
- The analysis on block size is thorough and well-explained.

**Weaknesses:**

- Section 3.1 (Video Tokenization) cannot be considered an original contribution, as the authors straightforwardly used MAGVITv2[1]. Labeling Section 3.1 as a preliminary section is recommended.
- Although the authors differentiate NBP from MAR[2] in Section 2, there is no supporting evidence that NBP offers denser supervised signals or greater training efficiency. To strengthen the paper’s contribution, it would help to include a comparison showing NBP’s advantage over MAR’s next set-of-tokens prediction by excluding the mask tokens.


[1] Yu, et al. "Language Model Beats Diffusion--Tokenizer is Key to Visual Generation." arXiv 2023.\
[2] Li, et al. "Autoregressive Image Generation without Vector Quantization." arXiv 2024.

**Questions:**

- The paper states that the model was trained on 17-frame videos, but the TATS score refers to a model trained on 16 frames. Could the authors clarify the process for measuring FVD with NBP models? Specifically, how is the first frame provided to the NBP model in UCI and K600 experiments, and is this frame included when measuring FVD?

- Should the blocks follow a raster scan order? While the authors state that the AR model’s unidirectional raster-scan pattern limits performance, NBP still uses this order in block-level. If it is not, extending the block size analysis in Section 4.5 to examine different block shapes could be beneficial. For example, each block could be constructed from non-nearby tokens within a clip or even from tokens across multiple clips.

### Suggestions
This paper could be more impactful by focusing on NBP's advantages over MAR [1] and providing a more comprehensive analysis of block design.

[1] Li, et al. "Autoregressive Image Generation without Vector Quantization." arXiv 2024.

---

> ### Author Response · Authors · 2024-11-25
> **Rebuttal by Authors**
>
> Thank you for the valuable comments.
>
> > W1: Section 3.1 (Video Tokenization) cannot be considered an original contribution
>
> R1: We will label Section 3.1 as a preliminary section in our revision, thanks for your advice.
>
> > W2:  There is no supporting evidence that NBP offers denser supervised signals or greater training efficiency than MAR
>
> R2: MAR employs a training approach similar to BERT/MaskGit, while our NBP follows the GPT style. **The GPT style inherently offers denser supervised signals and greater training efficiency than the BERT/MaskGit style**. Specifically, in MAR, 70%[1] of the full image token map is masked, requiring the model to recover these masked tokens based solely on the remaining 30% of unmasked tokens. Consequently, during each training step, **30% of the unmasked tokens do not receive any supervised signals, which diminishes training efficiency.**
>
> In contrast, **NBP does not utilize a masking mechanism, ensuring that 100% of the tokens receive supervised signals**. This enhancement in training efficiency is crucial for scaling up model training with larger model sizes and data volumes, similar to the improvements seen when transitioning from BERT to GPT.
>
> [1] See Sec 4.2 in MAR. It has a half probability of using 70% mask and another half probability of using 100% mask.
>
> > Q1: which part of frames are included when calculating FVD?
>
> A1: We follow the official setting (MAGVIT-v2, VideoPoet, OmniTokenizer, etc.) for training and evaluation. Specifically, we train on 17-frame videos. During inference, we generate 17-frame videos, then pick the first 16 frames for calculating metrics.
>
> > Q2: Should the blocks follow a raster scan order? Extending the block size analysis in Section 4.5 to examine different block shapes could be beneficial
>
> A2: Actually, the blocks in our framework follow the **vertical scan order** instead of the raster scan order. Specifically, the video token map size is (TxHxW)=(5x16x16), and we have investigated the block shape of (1x1x16), (1x4x16), (1x16x16) in our paper, which means the width of the block matches the width of the video token map, thus there are no cross-row interruptions.
>
> We also explore the performance of various block shapes (K600, 700M model) the results are as below:
> | Block Size | Block Shape (TxHxW) | FVD  |
> |:----------:|:-----------:|:----:|
> | 16         | 1x1x16      | **25.5** |
> | 16         | 1x4x4       | 33.4 |
> | 16         | 2x1x8       | 29.2 |
> | 8          | 1x1x8       | **25.7** |
> | 8          | 2x2x2       | 32.7 |
>
> Our findings indicate that **the official block shape of TxHxW=1x1x16 (generating row by row) outperforms other tested shapes such as 1x4x4 and 2x1x8**. We attribute this to two main factors:
>
> **(1) Token Relationships Within a Single Block**: The 1x1x16 block shape allows tokens within the block to represent a complete, continuous row, maintaining integrity without cross-row interruptions. In contrast, block shapes like 1x4x4 and 2x1x8 involve generating complex relationships across multiple rows and columns—or even frames—on a smaller spatial scale, posing greater challenges.
>
> **(2) Relationships Between Blocks**: The 1x1x16 block shape simplifies the modeling process to primarily vertical relationships between rows, which enhances continuity and consistency during the generation process, thereby reducing breaks and error accumulation.
>
> Additionally, we would like to **clarify our rationale for NOT using blocks that span multiple frames (block shape with T>1) in our initial design**:
>
> **(1) Temporal Compression**: Given that input videos are sampled at 8 fps/16 fps and undergo 4x temporal downsampling during tokenization, there is already a high level of information compression in the temporal dimension. Modeling rapidly changing content simultaneously across frames proves challenging.
>
> **(2) Causality in Temporal Modeling**: Our goal is for the NBP framework not only to excel in video generation but also to serve as a potential world model. Since video represents the world in spatiotemporal dimensions and temporal changes in the world are causal, we aim to **preserve complete causality in the temporal dimension during video generation**. Block shape with T=1 avoids introducing bidirectional temporal attention, aligning with our philosophy of using an auto-regressive generator (a decoder-only transformer) and a tokenizer like MagVITv2 with T=1 as the temporal unit.

---

> > ### Comment · Reviewer_G3HK · 2024-12-01
> >
> > First, I'd like to appreciate the authors for responding to the review thoroughly.
> >
> > The concern about the difference in supervised signals between NBP and MAR has been resolved. The authors also provided extra experiments and insights about designing the shape of the blocks, which I think are valuable.
> >
> > For Q1, I also read the discussions from other reviewers and the authors' answers. From what I understand, in UCF-101, for NBP model, the GT frame is always included as the first frame for FVD measurement. I think including the GT frame while measuring the FVD might boost up the FVD score of NBP and the authors should clarify this. Since the authors did not provide scores based on the 16 frames followed by the first GT frame, I cannot confirm if the proposed NBP really performs better than other models.
> >
> > Because of this, I think there is not enough evidence to fully support the paper’s main claim and will keep my original score.

---

### Official Review · Reviewer_vjce · 2024-11-04

**Soundness:** 1
**Presentation:** 2
**Contribution:** 2
**Rating:** 3
**Confidence:** 5

**Summary:**

This paper proposes next-block prediction framework as a semi-autoregressive method, enhancing the spatiotemporal integrity and parallel prediction for video generation tasks. Several modifications including initial condition, block-wise attention and inference process are applied to existing AR models, and massive experiments are conducted to find the optimal configuration of the block size. The proposed model reaches leading performance compared to previous SOTAs with a good scaling-up law.

**Strengths:**

- The proposed block-wise semi-AR method is novel and illustrated clearly.

- Rich comparisons and ablations with visualizations are presented and analysed.

**Weaknesses:**

- [Major] Line 370-372 mentions that the proposed method is first-frame conditioned, which is significantly different from other methods' settings (class-conditioned generation) in Table 3. This indicates completely unfair comparisons.

- The ablations on block size are not fine-grained enough given that the optimal point is 16. Additional values in [1, 16] and [16, 64] should be also investigated. Besides, what is the best block size for temporal axis? 1 is used for all experiments without discussion.

**Questions:**

- Is the tokenizer completely identical to MAGVIT-v2 or is there any modifications? Its performance is tested separately in Table 1, but the paper describes its architecture as the same as MAGVIT-v2. Also, in Table 1 the reconstruction performance falls behind vanilla MAGVIT-v2 with comparable parameter size.

---

> ### Author Response · Authors · 2024-11-25
> **Rebuttal by Authors**
>
> Thank you for the valuable comments.
>
> > W1: the proposed method for class-conditioned generation (UCF) is first-frame conditioned, which is different from other methods' settings in Table 3.
>
> R1: We understand the concern regarding our framework's dependency on an existing initial frame. To clarify the impact of this first-frame condition, we evaluate the performance of OmniTokenizer (previous SOTA auto-regressive baseline) on UCF:
> | Model                                 | Model Size | FVD   |
> |-------------------------------------|:----------:|:-----:|
> | OmniTokenizer  (w/o first frame condition)  | 650M       | 191.0 |
> | OmniTokenizer (w/ first frame condition) | 650M       | 133.2 |
> | NBP (w/ first frame condition)           | 700M       | **55.0**  |
>
> As shown in the above table, even when adjusted to a first-frame condition, the OmniTokenizer achieves an FVD of 133.2, which is still far behind NBP with 55.0 FVD.
>
> We acknowledge the limitation imposed by the initial frame requirement and have explored alternatives such as introducing a special token block (e.g., [begin_of_img]) to initiate generation from class names alone. However, this approach struggled with transitioning from the special tokens to generating meaningful content, resulting in suboptimal performance. Consequently, we did not pursue this method in our current work.
>
> In response to this issue, we are considering a hybrid method that could initially focus on next-token prediction and then transition to next-block prediction. We leave it for future work.
>
> Additionally, it is important to note that on the K600, all models were evaluated under uniform conditions. Here, our NBP framework consistently outperformed existing autoregressive models, which verifies the superiority of our framework.
>
> > W2: The ablations on block size are not fine-grained enough given that the optimal point is 16. Besides, what is the best block size for the temporal axis?
>
> R2: We explore the performance of more fine-grained block sizes (1x1x8 and 1x1x32) on K600, using the 700M model; the results are as below:
> | Block Size | Block Shape (TxHxW) | FVD  |
> |:----------:|:-----------:|:----:|
> | 1   | 1x1x1  | 29.0 |
> | 8   | 1x1x8  | 25.7 |
> | 16  | 1x1x16 | **25.5** |
> | 32 | 1x1x32 | 32.1 |
> | 64  | 1x1x64 | 43.7 |
>
> **Our findings confirm that a block size of 16 (1x1x16) remains optimal**. A block size of 16 indicates generating row by row, which helps maintain a high degree of continuity and consistency during the generation process, reducing breaks and error accumulation.
>
> Further, we investigated different block shapes along the temporal axis:
> | Block Size | Block Shape (TxHxW) | FVD  |
> |:----------:|:-----------:|:----:|
> | 16         | 1x1x16      | **25.5** |
> | 16   | 2x1x8       | 29.2 |
> | 8   | 1x1x8       | **25.7** |
> | 8    | 2x2x2       | 32.7 |
>
> The results show that block shapes with **T=1 generally perform better**. This finding **supports our initial design decision NOT to use blocks spanning multiple frames** for two main reasons:
>
> **(1) Temporal Compression**: Given that input videos are sampled at 8 fps/16 fps and undergo 4x temporal downsampling during tokenization, there is already a high level of information compression in the temporal dimension. Modeling rapidly changing content simultaneously across frames proves challenging.
>
> **(2) Causality in Temporal Modeling**: Our goal is for the NBP framework not only to excel in video generation but also to serve as a potential world model. Since video represents the world in spatiotemporal dimensions and temporal changes in the world are causal, we aim to **preserve complete causality in the temporal dimension during video generation**. Block shape with T=1 avoids introducing bidirectional temporal attention, aligning with our philosophy of using an auto-regressive generator (a decoder-only transformer) and a tokenizer like MagVITv2 with T=1 as the temporal unit.
>
> In conclusion, our choice of using blocks with T=1 is strategic, aiming to effectively model the world's spatiotemporal dynamics while maintaining the causality of temporal changes.
>
> > Q1: Is the tokenizer completely identical to MAGVIT-v2 or is there any modifications?
>
> A1: As mentioned in Section 4.1, our implementation of the MAGVIT-v2 tokenizer includes several modifications from that in the official paper:
>
> **(1)** We replaced the LFQ with the FSQ quantizer due to its simplicity and fewer loss functions and hyper-parameters involved.
>
> **(2)** Instead of using StyleGAN, we opted for PatchGAN to improve training stability.
>
> **(3)** Limited by computational resources, we did not pre-train our model on ImageNet or use a large visual vocabulary. Our tokenizer uses a 64K vocabulary compared to the 262K used in the original MAGVITv2, which may slightly affect our results.
>
> Nevertheless, we note that the primary objective of this paper is to validate the semi-AR framework, rather than to achieve state-of-the-art tokenizer performance.

---

> > ### Comment · Reviewer_vjce · 2024-11-27
> >
> > Thank the authors for the response. It addresses my concern on the design of temporal block dimension, while however the major issue of non-standard benchmark settings (additional initial frame condition) still remains.
> >
> > The authors provide that one of the comparing methods has suboptimal performance on the image-to-video generation task. However, as the most general benchmark, nosie-to-video generation task has been utilized to measure almost all of the relate work. Lack of good results on this benchmark projects the concern of the generalization and universality of the proposed approach, i.e. it highly relies on a good initialization of the beginning block token otherwise it fails to produce competitive output. A suggestion is to develop additional modules to produce a good initialization of the beginning block token so that the whole pipeline serves as a general one, if this dependency is inherent in the block prediction paradigm. Meanwhile, I also have doubts on the reasons provided for this issue, since both next token prediction and next frame prediction can sample from random noise as the beginning, why next block prediction, as a hybrid method of them, significantly underperform on this and is even not comparable?
> >
> > On the other hand, if image-to-video generation is mainly pursued in this paper, related work and comparing methods should be introduced and compared. Although the authors provide a new table in the discussion, it only contains one baseline and all other comparing methods are not tested for this non-standard task. Nonetheless, Table 3 etc. in the original PDF remains unchanged with the original unfair comparison, i.e. the proposed frame-conditioned generation vs. others' unconditional generation. Such presentation and illustration is still very misleading.
> >
> > Overall, I still maintain my opinion toward rejection.
> >
> > By the way, since there is no official MAGVIT-v2 codebased released yet, and there is no public video MAGVIT-v2 codebase yet (the only public one is for images), as far as I know, it still remains unclear how the proposed method is built.

---

### Official Review · Reviewer_gTpp · 2024-11-10

**Soundness:** 2
**Presentation:** 3
**Contribution:** 2
**Rating:** 3
**Confidence:** 5

**Summary:**

This paper proposes a semi-autoregressive (semi-AR) framework, Next-Block Prediction (NBP), for video generation tasks. Compared to the conventional autoregressive (AR) framework, or Next-Token Prediction (NTP), the proposed framework generates blocks with multiple tokens, while these blocks follow a raster-scan ordering. Thus, NBP reduces the number of forward steps required for sampling videos. Experimental results demonstrate that NBP can achieve low FVD scores on UCF-101 and K600 datasets.

**Strengths:**

S1. This paper aims to resolve an important issue in video generation, sampling efficiency.

S2. This study shows that a semi-AR framework, which is unexplored in video generation tasks, can also be used for video generation.

**Weaknesses:**

W1. Limited novelty and originality. Contrary to the claims in Section 2, it is widely known that the conventional semi-AR semi-AR framework predicts multiple tokens without additional modules. For example, the SAT model [NewRef-1], which is well-known and presented at EMNLP’18, also shares the same framework as the proposed approach. Thus, I believe the contribution of this paper does not lie in the framework design itself, but lie in applying existing semi-AR frameworks from NLP domains to video generation.

W2. Lack of in-depth analysis on the proposed block predictions. The ablation study does not explore various block shapes (e.g., 1x4x4). Especially, despite the video generation framework, there is no experiment involving the prediction of multiple tokens across different frames.

W3. Given that the proposed NBP conducts row-by-row generation, the framework should be validated on image generation tasks first. Note that the proposed transformer lacks a tailored design for video data.


[NewRef-1] Wang et al., Semi-Autoregressive Neural Machine Translation, EMNLP2018.

**Questions:**

Q1. Could the authors provide a more detailed explanation and comparison regarding sampling costs? Was KV-caching used in this comparison? Given that the FLOPs for both NBP and NTP are likely similar for sampling, I believe the inference speed should be comparable when using KV-caching as the model scales, even though NBP requires fewer forward steps than NTP.

Q2. Given the same block size, how does performance vary according to block shape? For instance, the ablation study could include comparisons like (1x1x16 vs. 1x4x4 vs. 16x1x1) or (1x16x16 vs. 16x4x4 vs. 4x8x8). Since the authors claim that NTP cannot account for spatial dependencies in local tokens, I initially expected the study to use 2D or 3D shapes for local blocks. However, it employs a 1D block shape, which has fewer spatial dependencies than 2D or 3D blocks.

Q3. In Table 3, how were PSNR, SSIM, and LPIPS computed for the generation results when no ground-truth data exists for video generation?

Q4. In Figure 5, why do the validation loss curves exhibit noisy patterns? I suspect these might be training losses rather than validation losses, considering the curve shapes in Figure 8. Additionally, given the large number of trainable parameters and epochs relative to the small dataset (such as UCF-101), I wonder whether the model shows signs of overfitting.

Q5. Since the experiments focus primarily on class-conditional generation for UCF-101 and frame prediction for K600, could the authors clarify how the text tokens are utilized?

Minor comments (not affecting the score):
- Eq. (1) may contain an error. $x_{l}^{(<t)}​$ should likely be $x^{(<t)}$.
- Since FVD is an incomplete metric for video generation, I recommend including additional metrics such as IS, Dover-Scores, Frame-wise Text Alignments, etc.
- Contrary to the statement in Lines 355-356, Flash Attention does not support customized attention masks.

---

> ### Author Response · Authors · 2024-11-25
> **Rebuttal by Authors (1)**
>
> Thank you for the valuable comments.
> > W1: The proposed approach is similar to the conventional semi-AR framework in machine translation
>
> R1: We acknowledge that semi-autoregressive (semi-AR) is indeed a long-standing concept. However, the motivation and goal of our work is not to reinvent this concept, but to reevaluate and adapt the design of autoregressive (AR) generation frameworks specifically for video generation.
>
> Our work addresses several key questions: **(1)** Should the optimal generating unit in AR be a single token, or a block composed of multiple tokens? **(2)** In video generation, should the AR framework exclusively utilize causal attention or integrate both causal and bidirectional attention? **(3)** How can we mitigate the inefficiencies and risks of error propagation due to the extensive forward steps required during AR inference?
>
> In response to these considerations, we introduced the next-block prediction (NBP) framework, which incorporates elements of semi-AR. Our approach involves modifying the AR generating unit to a block, using bidirectional attention within blocks, and causal attention between blocks, which enhances performance (-4.4 FVD on K600) and significantly speeds up inference by 11 times.
>
> Moreover, our NBP framework preserves the scalability of traditional AR models, as well as the flexibility and simplicity for modality and task unification. This adaptation and its implications constitute the major contributions of our work. We will refine the statements in our revision to more accurately reflect and emphasize these contributions.
>
> > W2&Q2: The ablation study does not explore various block shapes (e.g., 1x4x4). there is no experiment involving the prediction of multiple tokens across different frames.
>
> R2&A2: We explore the performance of various block shapes on K600, using the 700M model, the results are as below:
> | Block Size | Block Shape (TxHxW) | FVD  |
> |:----------:|:-----------:|:----:|
> | 16         | 1x1x16      | **25.5** |
> | 16         | 1x4x4       | 33.4 |
> | 16         | 2x1x8       | 29.2 |
> | 8          | 1x1x8       | **25.7** |
> | 8          | 2x2x2       | 32.7 |
>
> Our findings indicate that **the official block shape of TxHxW=1x1x16 (generating row by row) outperforms other tested shapes such as 1x4x4 and 2x1x8**. We attribute this to two main factors:
>
> **(1) Token Relationships within a Single Block**: The 1x1x16 block shape allows tokens within the block to represent a complete, continuous row, maintaining integrity without cross-row interruptions. In contrast, block shapes like 1x4x4 and 2x1x8 involve generating complex relationships across multiple rows and columns—or even frames—on a smaller spatial scale, posing greater challenges.
>
> **(2) Relationships between Blocks**: The 1x1x16 block shape simplifies the modeling process to primarily vertical relationships between rows, which enhances continuity and consistency during the generation process, thereby reducing breaks and error accumulation.
>
> Additionally, we would like to **clarify our rationale for NOT using blocks that span multiple frames (block shape with T>1) in our initial design**:
>
> **(1) Temporal Compression**: Given that input videos are sampled at 8 fps/16 fps and undergo 4x temporal downsampling during tokenization, there is already a high level of information compression in the temporal dimension. Modeling rapidly changing content simultaneously across frames proves challenging.
>
> **(2) Causality in Temporal Modeling**: Our goal is for the NBP framework not only to excel in video generation but also to serve as a potential world model. Since video represents the world in spatiotemporal dimensions and temporal changes in the world are causal, we aim to **preserve complete causality in the temporal dimension during video generation**. Block shape with T=1 avoids introducing bidirectional temporal attention, aligning with our philosophy of using an auto-regressive generator (a decoder-only transformer) and a tokenizer like MagVITv2 with T=1 as the temporal unit.
>
> > W3: The framework should be validated on the image generation task first.
>
> R3: As discussed in Section 3.3 (1) on Initial Conditions, our framework is designed to start with an existing initial frame for generating subsequent blocks, making it unsuitable for image generation where no analogous "first frame" exists.
>
> We attempted to adapt the framework by introducing a special token block (e.g., [begin_of_img]) as an initial condition (thus supporting the generation of images or videos from scratch). However, this approach struggled with transitioning from the special tokens to generating meaningful content, resulting in suboptimal performance. Consequently, we did not pursue this method in our current work.
>
> In response to this issue, we are considering a hybrid method that could initially focus on next-token prediction and then transition to next-block prediction. We leave it for future work.

---

> ### Author Response · Authors · 2024-11-25
> **Rebuttal by Authors (2)**
>
> > Q1: Was KV-caching used in this comparison? Given that the FLOPs for both NBP and NTP are likely similar for sampling, I believe the inference speed should be comparable when using KV-caching as the model scales, even though NBP requires fewer forward steps than NTP.
>
> A1: Yes, KV-cache is implemented in the inference and speed measurements for both NTP and NBP.
>
> In scenarios utilizing KV-Cache, the overall computation cost during each inference step for NTP involves multiplying vectors (current token) with matrices (model weights), which is primarily **IO-bound** due to the movement of matrices. Conversely, in the NBP model, the computation involves multiplying matrices (current block) with matrices (model weights), making it **compute-bound**, with reduced IO overhead due to larger block sizes. Given this distinction and assuming adequate GPU parallelism, the NBP framework can achieve significantly faster speeds compared to NTP. This efficiency gain is due to the reduced frequency of IO operations and the more effective utilization of computational resources in processing larger data blocks simultaneously.
>
> > Q3: How were PSNR, SSIM, and LPIPS computed for the generation results when no ground-truth data exists for video generation?
>
> A3: For K600 (frame prediction task), the official evaluation setting is to take the first 5 frames from the val videos as a condition and then predict the successive 12 frames; thus, we can take the successive 12 frames as ground-truth for PSNR, SSIM, and LPIPS calculation. For UCF (class-conditional generation task), we take the first frame from training samples as the condition, and take the successive 16 frames from training samples as ground-truth
>
> > Q4: In Figure 5, why do the validation loss curves exhibit noisy patterns? I wonder whether the model shows signs of overfitting for the small dataset like UCF
>
> A4: The noisy patterns observed for the 700M model are due to the smaller log interval. Specifically, the log interval for the 700M model is 6660, compared to 13319 for the 1.2B/3B models.
>
> Regarding the UCF dataset, we adhered to the official training and evaluation protocol (detailed in Appendix A), which includes 9.5K training data and uses training videos to approximate the real distribution for evaluation. It is recognized that smaller datasets like UCF can result in high variance. Consequently, our primary experimental focus, as presented in Table 2, has been on the K600 dataset, which is much larger and more reliable with 384K training data and 29K validation videos representing the real distribution for evaluation.
>
> > Q5: How the text tokens are utilized?
>
> A5: In our NBP, text tokens of class names are used as the condition for video generation. For instance, the UCF-101 dataset includes 101 classes. Traditionally, each class is represented by a unique token (token IDs range from 0 to 100). However, in our approach, we convert each class name into text tokens using the LLaMA tokenizer. For example, the class name 'apply eye makeup' is tokenized into [1, 3394, 10977, 1207, 786]. This method of tokenization facilitates the adaptation of our codebase for future text-to-video generation applications.
>
> > Minor comments: Contrary to the statement in Lines 355-356, Flash Attention does not support customized attention masks.
>
> Thanks for your other comments. We will revise our paper as suggested. For flash attention, it is supported by the function `torch.nn.functional.scaled_dot_product_attention`, which allows for customized attention masks in Flash Attention 2.
>
> https://pytorch.org/docs/stable/generated/torch.nn.functional.scaled_dot_product_attention.html

---

> ### Comment · Reviewer_gTpp · 2024-12-01
>
> I appreciate the authors' responses. Unfortunately, we could not make enough discussion due to the authors' late submission of responses. I thoroughly read the authors' response, but I decided to keep maintaining my initial scores.
>
> 1. On the motivation and the originality of this paper.
>
> > R1: We acknowledge that semi-autoregressive (semi-AR) is indeed a long-standing concept. However, the motivation and goal of our work is not to reinvent this concept, but to reevaluate and adapt the design of autoregressive (AR) generation frameworks specifically for video generation.
>
> I cannot agree with this response. If the authors aimed not to reinvent the semi-AR concept, but to reevaluate and adapt the design of AR for video generation, the authors should have already analyze the two primary things thoroughly: 1) the previous concepts in semi-AR for NLP, which is already invented and successfully applied to various approaches, 2) the special consideration for videos, where its speciality lies in spatiotemporal relationships between pixels. However, as also mentioned in the other reviewer's comments, this paper mainly lacks the two parts rather.
>
> 2. On the ablation study with various block size.
>
> Different from the claims/motivations in the paper, the ablation study's results show that the effectiveness of semi-AR for image/video generation should aim to learn non-correlated pixels/latents in parallel, instead of considering the spatial-temporal relationship between pixels/latents. Based on these results, the paper should be re-written to make it have more logical flow.
>
>
> 3. I/O bound of Speed Measurement.
> I couldn't find the detailed comparison to clarify the authors' claim that the I/O bound could make the presented differences in generation speed. Even though it is the major reason for the generation speed difference, the authors should clarify that the generation FLOPs between NTP and NBP are almost similar, but the difference comes from the I/O bound.
>
> 4. Experimental Results
> - Q3: I think the paper should be improved to clarify the presentation of results.
> - Q4: I'm still wondering how the 2x increased logging frequency makes such the differences in training stability. I'm also thinking that the curve is the training curve, not the validation curve, based on Figure 8.
>
>
>
> 5. On the Minor Comment
> > For flash attention, it is supported by the function torch.nn.functional.scaled_dot_product_attention, which allows for customized attention masks in Flash Attention 2.
>
> It's not. `torch.nn.functional.scaled_dot_product_attention` supports various backends, but FlashAttention2 does not support the custom attention mask. If a user specifies the backend to only be FlashAttention and gives attention_mask, the program returns error. If the authors allow various backend, pytorch automatically change the backend to leverage attention_mask.

---

### Meta-Review · Area_Chair_cr3m · 2024-12-18

**Metareview:**

This paper introduces a semi-autoregressive approach to video generation, shifting from next-token to next-block prediction. By grouping multiple tokens into blocks (e.g., row-wise), the proposed model can predict multiple tokens in parallel per step. The authors use bidirectional attention within each block and causal attention between blocks. They claim this approach achieves both significantly faster inference and improved generation quality compared to traditional autoregressive methods.

However, the reviewers uniformly expressed concerns and ultimately recommended rejection. Their primary criticisms include:

1. **Limited Novelty:** Semi-autoregressive and blockwise prediction strategies have been widely explored in NLP and, to some extent, vision tasks. The reviewers found the contributions incremental and insufficiently distinct from existing work.
2. **Unfair Comparisons in Evaluation:** The proposed method conditions on an initial ground-truth frame, a setting that differs from standard baselines (e.g., unconditional or class-conditional generation from scratch). This discrepancy raises doubts about the fairness and rigor of the reported performance gains.
3. **Insufficient Video-Specific Considerations:** While pitched as a video generation framework, the method could easily be applied to images and does not convincingly incorporate video-specific inductive biases. The authors do not provide a strong argument or evidence that their blockwise approach is particularly suited to modeling temporal aspects of videos.

In their rebuttal, the authors attempted to address these concerns, offering clarifications on their claimed novelty and presenting additional ablations. Unfortunately, these responses did not adequately satisfy the reviewers’ major concerns. All reviewers maintained their initial evaluations, agreeing that the paper lacks a sufficiently strong contribution beyond known semi-autoregressive techniques, relies on a potentially unfair evaluation setup, and does not convincingly justify its design choices for video data.

After carefully reviewing the submission, the reviews, and the author rebuttal, AC concurs with the reviewers’ assessment. The paper’s approach, while potentially of some practical interest, does not provide enough original insight or video-specific modeling advantages. Moreover, the evaluation methodology remains problematic due to the reliance on an initial ground-truth frame, which hinders a fair comparison to baselines. Consequently, AC recommends rejecting this submission at this time.

**Additional Comments On Reviewer Discussion:**

The three major weaknesses in the meta-review remained unresolved after the discussion.

---

### Decision · Program_Chairs · 2025-01-22

Reject